# α-Conotoxin as Potential to α7-nAChR Recombinant Expressed in *Escherichia coli*

**DOI:** 10.3390/md18080422

**Published:** 2020-08-12

**Authors:** Yanli Liu, Yifeng Yin, Yunyang Song, Kang Wang, Fanghui Wu, Hui Jiang

**Affiliations:** State Key Laboratory of NBC Protection for Civilian, Beijing 102205, China; liuh306@hotmail.com (Y.L.); shengfengfeiqu@sina.com (Y.Y.); syyfeng@126.com (Y.S.); yiyongjun1949@163.com (K.W.); wufanghui2050@163.com (F.W.)

**Keywords:** α-conotoxin ArIB (V11L; V16A), α7 nAChR, recombinant expression, electrophysiology

## Abstract

α7 nicotinic acetylcholine receptors (nAChR) is an important nicotinic acetylcholine receptors subtype and closely associated with cognitive disorders, such as Alzheimer’s and schizophrenia disease. The mutant ArIB (V11L, V16A) of α-conotoxin ArIB with 17-amino acid residues specifically targets α7 nAChR with no obvious effect on other nAChR subtypes. In the study, the synthetic gene encoding mature peptide of ArIB and mutant ArIB (V11L, V16A) carried a fusion protein Trx and 6 × His-tag was separately inserted in pET-32a (+) vector and transformed into *Escherichia coli* strain BL21(DE3) pLysS for expression. The expressions of Trx-ArIB-His_6_ and Trx-ArIB (V11L, V16A)-His_6_ were soluble in *Escherichia coli*, which were purified by Ni-NTA affinity chromatography column and cleaved by enterokinase to release rArIB and rArIB (V11L, V16A). Then, rArIB and rArIB (V11L, V16A) were purified by high-performance liquid chromatography (HPLC) and identified by matrix-assisted laser desorption ionization-time of flight mass spectrometry (MALDI-TOF MS). Bioactivity of rArIB and rArIB (V11L, V16A) was assessed by two-electrode voltage-clamp electrophysiology in *Xenopus laevis oocytes* expressing human nAChR subtypes. The results indicated that the yield of the fusion proteins was approximately 50 mg/L and rArIB (V11L, V16A) antagonized the α7 nAChR subtype selectively with 8-nM IC_50_. In summary, this study provides an efficient method to biosynthesize α-conotoxin ArIB and rArIB (V11L, V16A) in *Escherichia coli*, which could be economical to obtain massively bioactive disulfide-rich polypeptides at fast speed.

## 1. Introduction

Nicotinic acetylcholine receptors (nAChRs) are members of ligand-gated ion channels that belong to the Cys–loop receptor superfamily [1]. This superfamily also includes γ-aminobutyric acidA, the 5-HT3 serotonin, and the glycine receptors [2]. nAChRs are pentameric receptors formed ligand-gated ion channels by a combination of different or same subunits. As so far, 16 subunits of nAChRs have been confirmed that are α2–10, β1–4, δ, ε and γ subunits in mammals [3,4,5], and the homomeric (α7, α8, α9 or α10) or heteromeric (α2–α6 with β2–β4, α7β2 and α9α10) receptor complexes [6,7] produces many distinctive subtypes that have different pharmacological and physiological function [8].

Multiple neuronal nAChRs subtypes have been participated in the modulation of pain and immune system, such as the α4β2, α7 and α9α10 have received much attention in related fields, which are widely expressed in the brain. Among them, α4β2 and α7 nAChRs are confirmed to play a central role in nicotine addiction and in cognitive [9,10] and closely related with both psychiatric disorders (bipolar depression and schizophrenia) and neurological disorders (Parkinson’s disease, Alzheimer’s disease) [11]. The heteromeric, α9α10 nAChR subtype being expressed in outer hair cells, which is well-known for its role in the auditory system and has also been shown to be involved in cancer therapy [12]. The agents currently used for pharmacological recognition of α7 acetylcholine receptors are mainly α-Bgt and α-Cbt, which can also bind to α9* and α1* receptors co-expressed with α7 in various tissues and are poorly specific to make them more difficult to distinguish [13,14]. Therefore, Subtype-selective ligands that are capable of distinguishing multiple receptor subtypes are critical for studying the specific physiological effects of individual nAChRs.

α-Conotoxins is one of seven families of conotoxins known with pharmacological activity at nAChRs which is a kind of small disulfide-rich peptides containing 12–20 amino acids [15]. Some α-conotoxins exhibit high selectivity towards specific nAChR subtypes, for example, RgIA and Vc1.1 exhibit the analgesic effect to α9α10 and α9α10/α7 nAChRs in rat models of neuropathic pain, respectively [16,17,18,19]. There are studies also proving that Vc1.1 more potently inhibit N-type Cav 2.2 voltage-gated calcium channel currents (VGCC) via the G protein-coupled GABA_B_ receptor (GABA_B_R) to suppress pain in rat sensory neurons [20,21,22]. RegIIA chemically synthesized is a potent antagonist of the α3β4 nAChRs in *Xenopus laevis* oocytes, moreover, it can also activate α3β2 and α7 nAChRs [23]. α-Conotoxin EI mainly blocks α1β1δε nAChRs, applying the alanine scanning, the inhibition of EI(S13A) synthesized using solid phase methods is increased in a 2-fold at α1β1δε nAChR [24]. α-Conotoxin TxID highly targets α3β4 nAChRs [25], and it also exhibits the inhibition of α6/α3β4 nAChR, but TxID(S9K) selectively inhibited α3β4 nAChR [26].

α-conotoxins ArIB analogs ArIB (V11L, V16A) synthesized is a highly selective antagonist to α7 nAChR [27]. α-Conotoxin ArIB from *Conus arenatus* consists of 20-amino acid residues with the sequence of DECCSNPACRVNNPHVCRRR. It includes four cysteines (Cys), which form two couples of disulfide bonds with Cys (I–III) and Cys (II–IV) connectivity. The mutant ArIB (V11L, V16A) of α-conotoxin ArIB-labeled Cy3 was demonstrated to be >10,000-fold more selective for α7 nAChR with an IC_50_ of 2 nM over α9α10 and α1β1δε [28]. It can be used as a new tool to distinguish α7 from other nAChRs subtypes. Conotoxins from cone snails are difficult to obtain in large quantities, and large-scale capturing of cono snails will accelerate the destruction of the marine ecological environment. Chemical synthesis of conotoxins needs subsequent folding steps, which is still not economical and time consuming. Expression of soluble and functional peptides in *Escherichia coli* as alternative method has made a great progression, but it is still difficult, especially peptides with multiple disulfides such as conotoxins.

In this study, recombinant ArIB (rArIB) and mutant ArIB (V11L, V16A) was co-expressed with Thioredoxin (Trx) and His-tag as fusion protein in *Escherichia coli*. The results demonstrated that the high-yields of fusion protein rArIB and rArIB (V11L, V16A) were obtained at ≥50 mg/L, and the rArIB (V11L, V16A) selectively antagonized the human α7 nAChR subtype with an IC_50_ of 8 nM. The research reported an alternative approach to produce large quantity of bioactive disulfide-rich peptides in vitro was successfully established at low cost and fast speed, which will not only open up new methods for the development and utilization of conotoxins, but also have great significance to the protection of the marine ecological environment.

## 2. Results

### 2.1. Construction of pET–ArIB and pET–ArIB (V11L, V16A) Expression Vector

A pair of complementary oligonucleotide strands encoding α-CTx ArIB and ArIB (V11L, V16A) was designed and 5′-end phosphorylation synthesized, respectively using the preferred codon usage of *Escherichia coli* according to the mature peptide of α-CTx ArIB precursor gene (Table 1 and Figure 1), which were annealed to form double-stranded gene of rArIB and rArIB (V11L, V16A) and ligated with pET-32a(+) digested with *Kpn* I and *Eco*R I. The pET–ArIB and pET–ArIB (V11L, V16A) expression vector was successfully constructed (Figure 2). The recombinant fusion protein sequentially comprised a Trx protein and a His-tag, followed by a target peptide. Here, Trx as a fusion partner is a sequence of amino acids which directs the target protein to correctly folding and removed by enterokinase. His-tag was used for the purification of target proteins by Ni-NTA affinity column.

### 2.2. Expression and Purification of Recombinant Trx-His_6_-ArIB and Trx-His_6_-ArIB (V11L, V16A) in Escherichia coli

To express functional peptides in *Escherichia coli* and develop a system for producing cysteine-rich peptides, Trx was used to improve solubility of target peptides and correctly folded of disulfide bond as a fusion partner. Generally, following sequence verification, the constructs were transformed into BL21(DE3) pLysS of *Escherichia coli* to express fusion protein of TrxA-His**_6_**-ArIB and TrxA-His**_6_**-ArIB (V11L, V16A) under IPTG induction. As analyzed using 15% sodium dodecyl sulfate polyacrylamide gel electrophoresis (SDS-PAGE), bands with a molecular weight of ~20 kD was significantly increased compared with control after induced by 1-mM isopropyl β-D-1-thiogalactopyranoside (IPTG) for 4 h at 37 °C (Figure 3a, lane 3,5,7 and Figure 3b, lane 3,4). The soluble Trx-His**_6_**-ArIB and TrxA-His**_6_**-ArIB (V11L, V16A) was then subjected to centrifuged, sonicated, filtered and purified by HisPur^TM^ Ni-NTA purification kit. As analyzed using 15% SDS-PAGE (Figure 4a, lane 1–3 and Figure 4b, lane 1–4), the fusion proteins Trx-Fusion proteins Trx-His**_6_**-ArIB and TrxA-His**_6_**-ArIB (V11L, V16A) were eluted from Ni^2+^ column by 250-mM imidazole. The eluted proteins were subjected to ultrafiltration in order to remove the salt and concentrate the fusion proteins, the yield of fusion proteins was up to 50 mg/L.

### 2.3. Cleavage and Purification of Trx-His-ArIB and Trx-His-ArIB (V11L, V16A)

To obtain rArIB and rArIB (V11L, V16A), the fusion proteins were cleavage by enterokinase at 23 °C for 24 h and α-CTX rArIB and rArIB (V11L, V16A) were purified and analyzed by HPLC (Figure 5). The results showed that α-CTX rArIB and rArIB (V11L, V16A) were both purified successfully and eluted at about 50% solvent B with retention time of 22.52 ± 0.41 (Figure 5a) and 24.29 ± 0.30 (Figure 5b), individually. The molecular mass of α-CTX rArIB and rArIB (V11L, V16A) were confirmed by MALDI-TOF MS. The *m/z* (+H) of rArIB and rArIB (V11L, V16A) were, respectively 2325.146 and 2310.881(Figure 6a,b), which were consistent with the theoretical monoisotopic mass (2323.993 Da and 2309.978 Da) of folded peptide with two disulfide bonds.

### 2.4. Effect of rArIB and rArIB (V11L, V16A) on ACh-Evoked Current of nAChRs

The inhibition of ACh-induced hα7 responses in *Xenopus laevis* oocytes was used to assess the activity of both rArIB and rArIB (V11L, V16A). As shown in the Figure 7a,c, both rArIB and rArIB (V11L, V16A) exhibited inhibition on ACh-evoked current produced by hα7, of which, the IC_50_ values were 20 nM and 8 nM, respectively (Figure 7b,d). Fluorescein-labeled probes for α7 ACh receptor have so far been derived from α-Bgt and α-Cbt, however, α-Bgt and α-Cbt can also bind to the α9* and α1* receptors in various tissues and have no specificity. The inhibition of ACh-induced hα9α10, hα1β1δε responses in *Xenopus laevis* oocytes was evaluated the specificity of ArIB (V11L, V16A) on α7 ACh, while no obvious block on hα9α10, hα1β1δε nAChR subtypes at the concentration of 30 μM of ArIB (V11L, V16A) were observed (Figure 7e,f). Taken together, rArIB (V11L; V16A) can specifically antagonize the α7 nAChR.

## 3. Discussion

Conotoxins are a kind of small bioactive peptides ribosomally synthesized in cone snails that consist of 10–40 amino acids, include 2–4 disulfides [29]. Owing to the structural stability, target specificity and relatively small size, conotoxins are regarded as a rich source of molecular probes in neuroscience and have promising therapeutic applications. Many identified conotoxins have shown diverse physiological activities that target different receptors of the cell membrane including nAChRs, 5-hydroxytryptamine receptors (5-HT3Rs), N-methyl-D-aspartate receptors (NMDA) antagonists and α-amino-3-hydroxy-5-methyl-4-isoxazole propionate (AMPA) enhancers [30]. It is estimated that there are approximately 50,000 conotoxins could be secreted by different Conus species, as so far, although over 10,000 conotoxin sequences were published [29], only a little of them have been developed and utilized to pharmacological research and clinical application.

According to diversity of targets including voltage-gate ion channels (sodium, potassium and calcium) and ligand-gated ion channels (nicotine receptors and NMDA receptors), conotoxins can be classified into several superfamilies consisted of μ-, μO-, δ-, ω-, κ-, κA, κM, κI, γ-, α-, αA-, ρ-, ψ-, λ-, etc [31]. Among them, α-conotoxin has been identified to be specifically pharmacological properties targeting a variety of nAChRs [15]. α-ArIB analogs ArIB (V11L, V16A) synthesized has been identified a highly selective antagonist to α7 nAChR. Therefore, acquiring massive ArIB (V11L, V16A) peptide at low cost could help the application of ArIB (V11L, V16A) as a probe tool to distinguish α7 from other nAChRs subtypes.

As so far, there are three major methods to obtain conotoxins: (1) separation and extraction from cone snails; (2) solid-phase peptide synthesis; (3) recombinant expression in different expression system. Separation and extraction conotoxins from cone snails are difficult to obtain in large quantities which could not meet the further research. There are great difficulties in solid-phase peptide synthesis for conotoxins with more than 30 amino acids in length, moreover, solid-phase peptide synthesis of conotoxins needs subsequent folding steps, which is not economical and time consuming. Recombinant expression technology represents an alternative approach of producing mass, bioactive protein and peptides, *Escherichia coli* is the most common used prokaryotic expression system because of genetic information clearly and simple operations [26]. To solve the inclusion body of the protein heterologously expressed in the cytoplasm of *Escherichia coli* generally forms inclusion body, there has been made great progress in preparing of small peptide conotoxins mostly with disulfide-rich in soluble form making them more difficult to be effectively expressed, properly folded using soluble fusion proteins such as Trx, SUMO, PelB leader and maltose-binding protein (MBP). Using purified tags is helpful for the purification of recombinant protein, such as glutathione *S*-transferase (GST) and hexahistidine (His_6_), etc. To date, over 11 conotoxins (α-TxIA, TxIB, LvIA, M-superfamily lt16a, lt15a, Bt15a, Ec15a, Cap15a, Vx15a, Vr15a, lt7a, μO-MrVIB and ω-MVIIa, PrIIIE, etc.) have successfully been recombinantly expressed in *Escherichia coli* [32,33,34,35,36,37,38,39,40,41,42] and bioactive TxVIA has been produced in the yeast *Pichia pastoris* [43].

There are following advantages in this work: to obtain a stable and effective expression system of rArIB and rArIB (V11L, V16A), a synthesized gene of ArIB and ArIB (V11L, V16A) with preference codons of *Escherichia coli* were separately inserted into pET-32a (+) vector. The pET-32a (+) vector has several features that facilitate the expression and purification of small proteins or peptides, which was designed to co-expressed with a His-tag for purification and a fusion protein Trx that helps to improve solubility of target peptides and correctly folded of disulfide bond in the cytoplasm [40,41]. To improve the expression lever of recombinant peptides, 2 × YT medium containing more richer nutrients was used in this study. In the study, the yield of fusion proteins rArIB and rArIB (V11L, V16A) was up to 50 mg/L in flasks. There are some studies applying Trx as a fusion protein to recombinant express conotoxins, It17a and ω-MVIIa [40,41]. However, the yield of recombinant It17a (27 amino acids) and ω-MVIIa (25 amino acids) respectively is 6 mg/L and 40 mg/L, which are much lower than the yield of fusion protein rArIB (20 amino acids) and rArIB (V11L, V16A) (20 amino acids). The production of peptides make great progression, for example, tandem repeats of LvIA gene fragment are cloned into PET-31b (+) and the yield of recombinant was up to 100–500 mg/L, however, the Cys connectivity of LvIA need renaturation by air oxidation in the research and N-terminal Met residue generate by CNBr cleavage that may be lead the inhibition of rLvIA to decrease on rat and human α3β2 nAChRs subtype compared with native LvIA. At the same time, in order to maximize the yield of fusion proteins, different temperatures (18 °C, 37 °C), concentrations of IPTG (1 mM and 0.1 mM) and induce time (4 h and 24 h) were tried in the study. There is little difference in the yield of recombinant peptides induced by 1 mM IPTG at 37 °C for 4 h and 0.1 mM IPTG at 18 °C for 24 h, which was contrary to the results reported by Pi Ch [40]. In Pi ch research, the lower temperature and lower concentration of IPTG were more beneficial to improve the soluble expression of lt7a.As the fusion protein of Trx and His-tag would have tremendous effect on the structure and bioactivity of rArIB and rArIB (V11L, V16A), for small peptides, the introduction of additional amino acids may affect the activity, compared with other protease cleavage, SUMO, TEV, Factor Xa and thrombin, etc. [44], enterokinase cleavage was used as an efficient method in the research to release rArIB and rArIB (V11L, V16A) from fusion protein Trx-ArIB-His_6_ and Trx-ArIB(V11L, V16A)-His_6_ without additional amino acid because of its high efficiency and specificity sequence DDDDK and the cleavage site at K. The cleavage condition of enterokinase had effect on the cleavage efficiency, which including the temperature and the ration of molar (peptide: enterokinase). After several rounds of trial, on the reaction conditions that enterokinase was added the solution by the ratio of 1:0.0006% (*M*:*M*) at 23 °C for 24 h, the cleavage efficiency of enterokinase was up to 95%, the ultimate cleavage condition was determined, and the yield of purified peptide was up to 5 mg/L. Subsequently, the pharmacological property of rArIB and rArIB (V11L; V16A) was evaluated by electrophysiology. The rArIB and rArIB (V11L; V16A) both could inhibit the current of human α7 receptors stimulated by acetylcholine, that is, ArIB and ArIB (V11L; V16A) can inhibit the activity of α7 with IC_50_ values of 20-nM and 8 nM, respectively. Compared with the native ArIB and chemical synthesis rArIB (V11L; V16A), the activity of rArIB and rArIB (V11L; V16A) was separately 2- and 4-fold decrease in inhibition of rα7 nAChR, the reason may be that human nAChRs were used in the electrophysiology experiment. There was no obvious effect on human α9α10 and α1β1δε receptors of rArIB (V11L; V16A), which did not block the current of acetylcholine to stimulate rα9α10 and rα1β1δε nAChRs that lack α7 subtype. The results were consistent with the results reported by Hone A.J. [28].

In summary, the study described an efficient and feasible method to the biosynthesis of bioactive α-CTx ArIB and ArIB (V11L, V16A) using Trx as the fusion partner, enterokinase as the cleavage protease and the His-tag as purification tag in *Escherichia coli*. This results not only may provide an alternative method to obtain disulfide-rich small peptides in large quantity and at low cost, but also provide a convenient and fast way to perform the structure-function research of peptides and accelerated the development and utilization of natural products.

## 4. Materials and Methods

### 4.1. Strains, Plasmid Vectors and Reagents

The *Escherichia coli* strains DH5α and BL21(DE3) pLysS, as well as expression vector pET-32a (+), were purchased from Novagen (Darmstadt, Hesse, Germany). Restriction enzyme *Kpn* I, *Eco*R I and *Mlu* I, T4 DNA ligase, protein markers, protein HisPur^TM^ Ni-NTA purification kit, Fetal Bovine serum (FBS), penicillin, streptomycin, and cRNA mMESSAGE mMACHINE In Vitro Transcription Kit were purchased from Thermo Scientific (Waltham, MA, USA). enterokinase was purchased from New England Biolabs, Inc. (Ipswich, MA, USA). Vydac C18 columns (5 μm, 4.6 mm × 250 mm, 10 μm, 22 mm × 250 mm) were purchased from Grace (Deerfield, IL, USA). Acetonitrile (ACN, gradient grade for HPLC), Acetylcholine chloride (ACh) and other chemical reagents were all of analytical grade and purchased from Sigma-Aldrich (St. Louis, MO, USA).

### 4.2. Construction of pET–ArIB and pET–ArIB (V11L, V16A) Expression Vector

The gene of ArIB and ArIB (V11L, V16A) with preference codons of *Escherichia coli* was designed based on the mature peptide sequence of α-ArIB precursor DNA (NCBI: P0C8R2.1). DDDDK was added to the 5′-end of the ArIB and ArIB (V11L, V16A) gene, which would be the recognition site for enterokinase (EK) cleavage after recombinant expression (Figure 1). The forward and reverse chains of ArIB and mutant ArIB (V11L, V16A) gene used for expression were synthesized and phosphorylated by Nanjing Genscript Biologic Technology Limited Company (Nanjing, China) (Table 1). They were dissolved in double-distilled water (ddH_2_O) at the ratio of 1:1 (*n:n*), then denatured at the condition of 95 °C for 10 min, annealed by gradient cooling to 25 °C. The vector pET-32a (+) was digested with *Kpn* I and *Eco*R I restriction enzyme for linearization. The linearized vector and the ArIB and ArIB (V11L, V16A) gene were ligated by T4 DNA ligase to construct the pET–ArIB and pET–ArIB (V11L, V16A) expression vector, which was transformed into strain DH5α of *Escherichia coli* (Figure 1). The recombinant vector pET–ArIB and pET–ArIB (V11L, V16A) was identified by DNA sequencing.

### 4.3. Recombinant Expression, Purification and Identification of Fusion Protein Trx-His_6_-ArIB and Trx-His_6_-ArIB (V11L, V16A)

The recombinant plasmid pET–ArIB and pET–ArIB (V11L, V16A) was transformed into strain BL21(DE3) pLysS of *Escherichia coli* to express fusion protein of Trx-His**_6_**-ArIB and Trx-His**_6_**-ArIB (V11L, V16A). A recombinant single colony was picked to inoculate in the 2 × YT medium containing 100-µg/mL ampicillin, which were incubated overnight in a shaking incubator with 250 rpm at 37 °C. The overnight culture was diluted at a ratio of 1% (*v/v*) with fresh 2 × YT medium containing 100-µg/L ampicillin, which was incubated at 37 °C with 250 rpm until the OD_600_ reached 0.6–0.8. Then, isopropyl-β-D-1-thiogalactopyranoside (IPTG) was added to the medium with a final concentration of 1-mM to induce recombinant expression. After incubated at 37 °C with 250 rpm for 4 h, the bacteria were harvested by centrifuged at 4500 × *g* for 15 min. The bacteria pellet acquired from per liter culture was washed with 100 mL of 20-mM ice-cold Tris HCl (pH 8.0) and resuspended in 80 mL ice-cold Lysis buffer of 20-mM Tris HCl (pH 8.0), 0.5-M NaCl. After repeated freezing and thawing three times, the solution was added lysozyme with a final concentration of 1 mg/mL. Then sonicated at 300 W for 60 cycles (10 s a cycle with 5 s working) in an ice–water bath to crush cells, the solution was centrifuged at 10,000× *g* for 10 min. The supernatant containing recombinant protein TrxA-His**_6_**-ArIB and TrxA-His**_6_**-ArIB (V11L, V16A) was analyzed by 15% SDS-PAGE.

Subsequently, the above supernatant was used for purification. After being centrifuged and filtered through the 0.45-μm filter, the supernatant with TrxA-His**_6_**-ArIB and Trx-His**_6_**-ArIB (V11L, V16A) was purified by HisPur^TM^ Ni-NTA purification kit (Thermo Scientific, Walthalm, MA, USA). The procedure for fusion protein purification was according to the Ni-NTA purification kit manufacturer’s instruction. Three incremental concentrations of 125, 250, and 500 mm of imidazole were used to elute recombinant fusion protein of Trx-His-ArIB and Trx-His-ArIB (V11L, V16A). All the eluted fractions were collected and analyzed by SDS-PAGE. The purified Trx-His-ArIB and Trx-His-ArIB (V11L, V16A) solution was desalted in 10 kD Vivaspin ultrafiltration spin columns at 12,000× *g* for 30 min, then lyophilized for the next experiment.

### 4.4. Cleavage and Purification of Trx-His-ArIB and Trx-His-ArIB (V11L, V16A)

The fusion protein lyophilized was dissolved in enterokinase digestion buffer (100 mM NaCl, 4 mM CaCl_2_, 40 mM Tris HCl, pH 8.0) with the final concentration of 2 mg/mL, while enterokinase was added the solution by the ratio of 1:0.0006% (*M*:*M*) at 23 °C for 24 h. The products after enterokinase cleavage was resuspended in 0.1% TFA and centrifuged at 13,000× *g* for 10 min to get the supernatant containing final product of rArIB and rArIB (V11L, V16A) peptide. Then the partial supernatants were dissolved in 0.1% trifluoroacetic acid (TFA) and applied to HPLC (Agilent 1100, Agilent Technologies Inc, Los Angeles, CA, USA) with Vydac C18 analytical column (5 μm, 4.6 mm × 250 mm). The major peaks were collected and analyzed by MALDI-TOF MS. After verified the major peak of rArIB and rArIB (V11L, V16A), the rest supernatants dissolved in 0.1% TFA were on a Vydac C18 semipreparative column (10 μm, 22 mm × 250 mm) of HPLC for purification by collecting the verified peak of rArIB and rArIB (V11L, V16A). The peptide of elution was performed with a linear gradient of 0–50% solvent B over 50-min at a flow rate of 1 mL/min, where solvent B was 90% ACN and 10% ddH_2_O with 0.05% (*v*:*v*) TFA; solvent A was ddH_2_O. Absorbance was monitored at 214 nm.

### 4.5. Mass Spectrometry

For rArIB and rArIB (V11L, V16A), MALDI-TOF MS analysis was performed on an Ultraflex MALDI-TOF mass spectrometer (Bruker Daltonics, Billerica, MA, USA), equipped with a 50 Hz pulsed nitrogen laser (λ = 355 nm) and a 19KV accelerating voltage operated in reflectron, positive ion mode. The samples were prepared by mixing 1 μL peptide solution with 1 μL matrix (α-cyano-4-hydroxycinnamic acid) saturated solution in 0.1% TFA containing 30% ACN. Data collection and processing was respectively performed by FlexControl 2.4 and FlexAnalysis 2.4 (Bruker Daltonics, Billerica, MA, USA).

### 4.6. cRNA Preparation and Injection

Plasmids of cDNA clones encoding *h*α7, α9, α10, α1, β1, ε and δ nAChR subunits were linearized by corresponding restriction enzyme. Each linearized plasmid was used as template to synthesize cRNA using mMESSAGE mMACHINE in vitro transcription Kit (Thermo Scientific, Waltham, MA, USA). The concentration of each cRNA was determined by its absorbance at 260 nm, individually. The cRNAs of different nAChR subunits were combined and 50 nL (at least 5–10 ng) injected into each *Xenopus* oocytes as described previously within 48 h of harvest [45]. The oocytes injected were incubated in the ND96 buffer (96-mM NaCl, 2-mM KCl, 1.8-mM CaCl_2_, 1-mM MgCl_2_ and 5-mM HEPES; pH 7.2) containing antibiotics (100 U/mL penicillin, 100 U/mL streptomycin and 2.5-mM pyruvate sodium) at 18 °C for 48 h.

### 4.7. Voltage Clamp Recording

The membrane currents of oocytes were recorded using a two-electrode voltage clamp equipped with an Axon 900A amplifier (Axon Instruments, Inc., Union City, CA, USA) and exposed to Ach and toxins as described previously after injected for 2–5 days. Briefly, the voltage clamping, and current recording electrodes were pulled from borosilicate glass. The diameter of their tips was controlled in the range of 1–3 μm, and the resistance was about 1–2 MΩ when filled with 3 M KCl. The oocyte chamber consisting of a cylindrical groove (about 50 μL in volume) was perfused gravity at a rate of 2–4-mL/min with ND-96 buffer containing 0.1 mg/mL BSA and 1 μm atropine for all hα1β1δε, hα7 and hα9α10 nAChR subtypes. The oocyte was voltage-clamped at a holding potential of −70 mV, which was subjected to 2 s ACh pulse once per min during recording. The ACh concentration was 100 μm for hα7, hα1β1δγ and hα9α10 nAChR subtypes. Once per stable baseline was obtained, either ND-96 alone or containing different concentrations of rArIB and rArIB (V11L, V16A) was pre-incubated with oocyte for 5 min prior to the exposure of ACh pulses. All data were collected at slow 100 Hz frequency and filtered at 5 Hz. All recordings were conducted at room temperature. Measurements from 4 to 6 oocytes were conducted in each treatment group and all data were presented as the mean ± SD. Estimates of potency were obtained by fitting concentration response curves to the data by the equation: Inhibition = 1 / [1 + (IC_50_ / C_toxin_)^n^] with nonlinear regression analysis using IGOR Pro 6.0.1.0 (WaveMetrics, Inc., Oswego, NY, USA), where n is the Hill coefficient and IC_50_ is the antagonist concentration giving half-maximal response.

## Figures and Tables

**Figure 1 marinedrugs-18-00422-f001:**
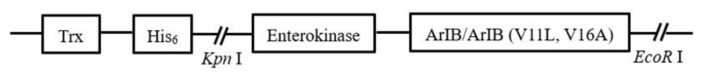
Construction of pET–ArIB and pET–ArIB (V11L, V16A) expression vector. The synthetic gene coding ArIB and ArIB (V11L, V16A) were inserted downstream of His6 tag between *Kpn* I and *Eco*R I sites of the expression vector pET-32a (+), respectively and enterokinase site was introduced at the N-terminal of ArIB and ArIB (V11L, V16A) for cleaving the Trx and His6-tag.

**Figure 2 marinedrugs-18-00422-f002:**
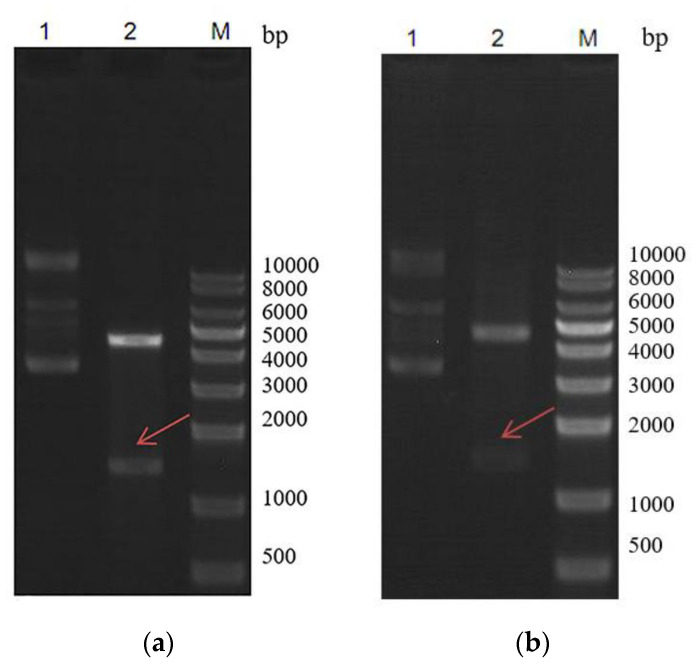
Gel image of recombinant plasmid pET–ArIB and pET–ArIB (V11L; V16A) for the expression of α-CTX ArIB and mutant ArIB (V11L; V16A) digested by Restriction enzyme. (**a**) Identification of recombination plasmid pET–ArIB digested by *Eco*R I and *Mlu* I; (**b**) identification of recombination plasmid pET–ArIBM digested by *Eco*R I and *Mlu* I digested by *Eco*R I and *Mlu* I.

**Figure 3 marinedrugs-18-00422-f003:**
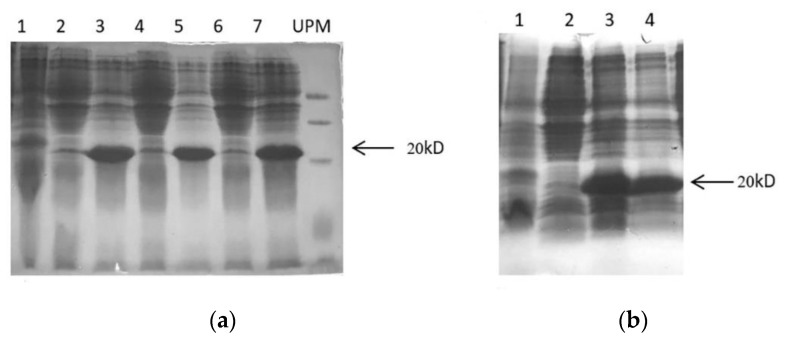
15% SDS-PAGE analysis of recombinant pET–ArIB and pET–ArIB (V11L; V16A) expressed in *Escherichia coli* BL21(DE3) pLysS cells. UPM: SpectraTM multicolor low range protein ladder 26628 (Thermo Scientific Waltham, MA, USA). (**a**) Recombinant expression of pET–ArIB induced by 1-mM IPTG at 37 °C. Lane 1, induced plasmid pET-32a (+) as negative control; Lane 2,4,6, uninduced recombinant plasmid pET–ArIB as positive control; Lane 3,5,7: induced recombinant plasmid pET–ArIB; (**b**) recombinant expression of pET–ArIB (V11L; V16A) induced by 1-mM IPTG at 37 °C. Lane 1, induced plasmid pET-32a (+) as negative control; Lane 2: uninduced recombinant plasmid pET–ArIB (V11L; V16A); Lane 3,4: induced recombinant plasmid pET–ArIB (V11L; V16A).

**Figure 4 marinedrugs-18-00422-f004:**
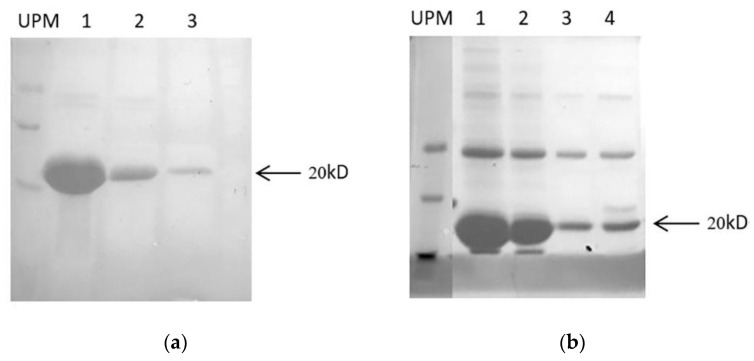
15% SDS-PAGE analysis of recombinant pET–ArIB and pET–ArIB (V11L; V16A) purified by HisPurTM Ni-NTA purification kit. UPM: SpectraTM multicolor low range protein ladder 26628 (Thermo Scientific, Waltham, MA, USA). (**a**) Purified Trx-His6-ArIB fusion protein, Lane 1–3: Eluted fractions after elution one times, 3 times and 5 times using 250-mM imidazole; (**b**) purified Trx-His6-ArIB (V11L; V16A) fusion protein, Lane 1–4: Eluted fractions after elution one time, three times, five times and six times using 250-mM imidazole.

**Figure 5 marinedrugs-18-00422-f005:**
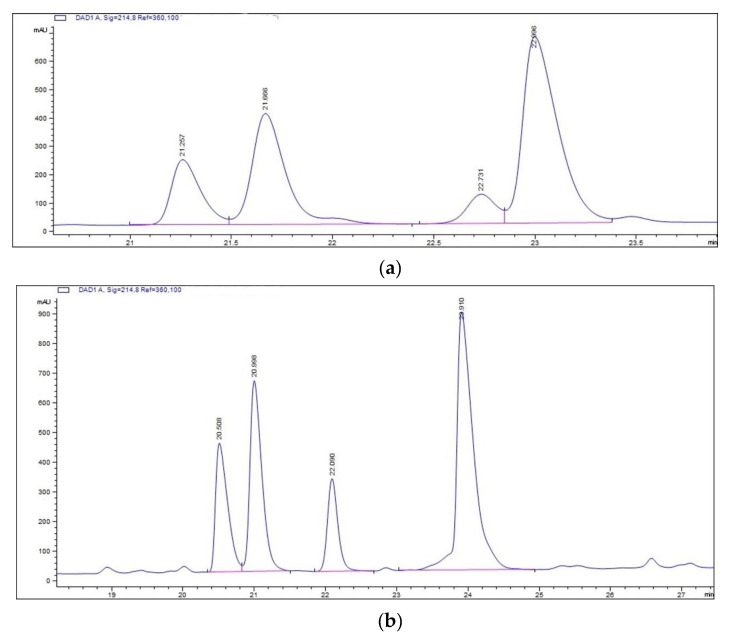
HPLC chromatogram of (**a**) purified rArIB and (**b**) rArIB (V11L; V16A). rArIB and rArIB (V11L; V16A) following enterokinase cleavage of thioredoxin was analyzed by HPLC with a Vydac C18 column (10 μm, 22 mm × 250 mm), using a linear gradient of 0–50% solvent B over 50-min at a flow rate of 1 mL/min, solvent B was 90% ACN and 10% ddH_2_O with 0.05% (*v:v*) trifluoroacetic acid (TFA), solvent A was ddH_2_O. Absorbance was monitored at 214 nm.

**Figure 6 marinedrugs-18-00422-f006:**
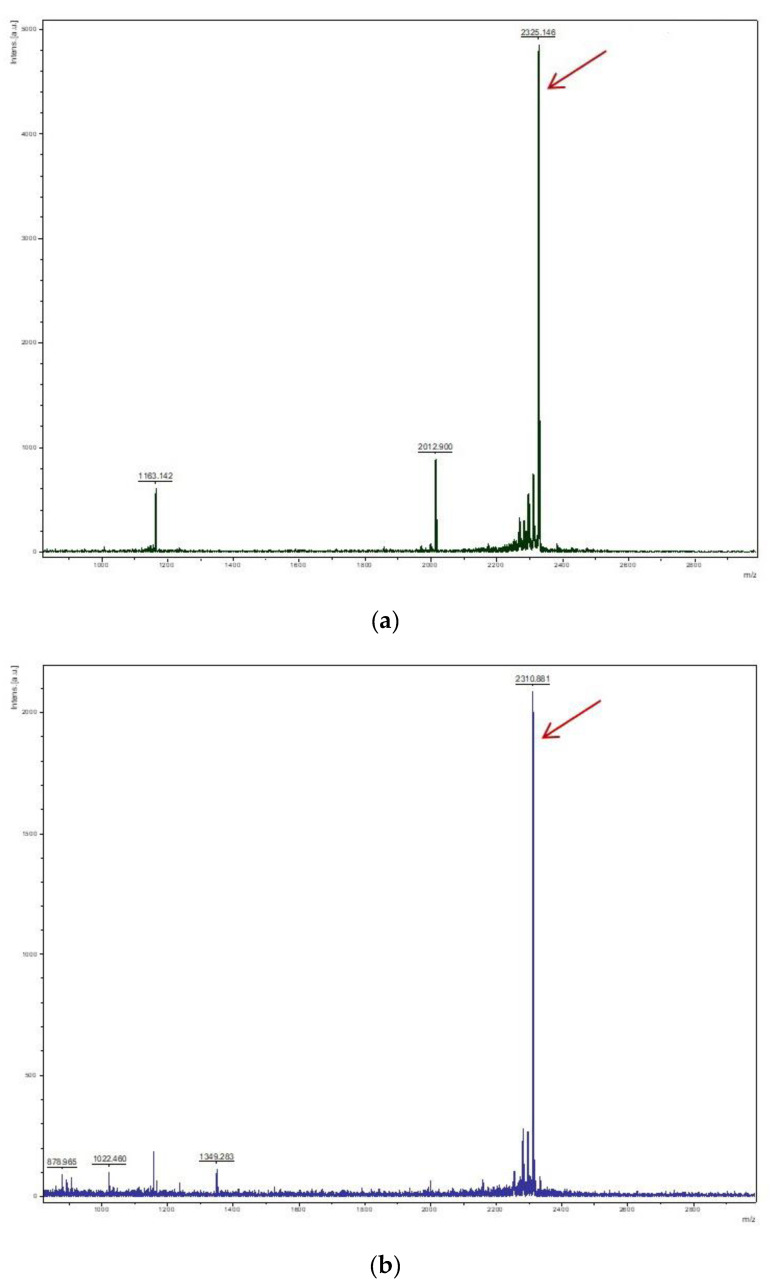
MALDI-TOF MS analysis of purified rArIB (**a**) and rArIB (V11L; V16A) (**b**); The *m/z* (+H) of rArIB (**a**) and rArIB (V11L; V16A) (**b**) with observed molecular weight of 2325.146 Da and 2310.881 Da, which were consistent with calculated theoretical monoisotopic mass of 2323.993 Da and 2309.978 Da, respectively.

**Figure 7 marinedrugs-18-00422-f007:**
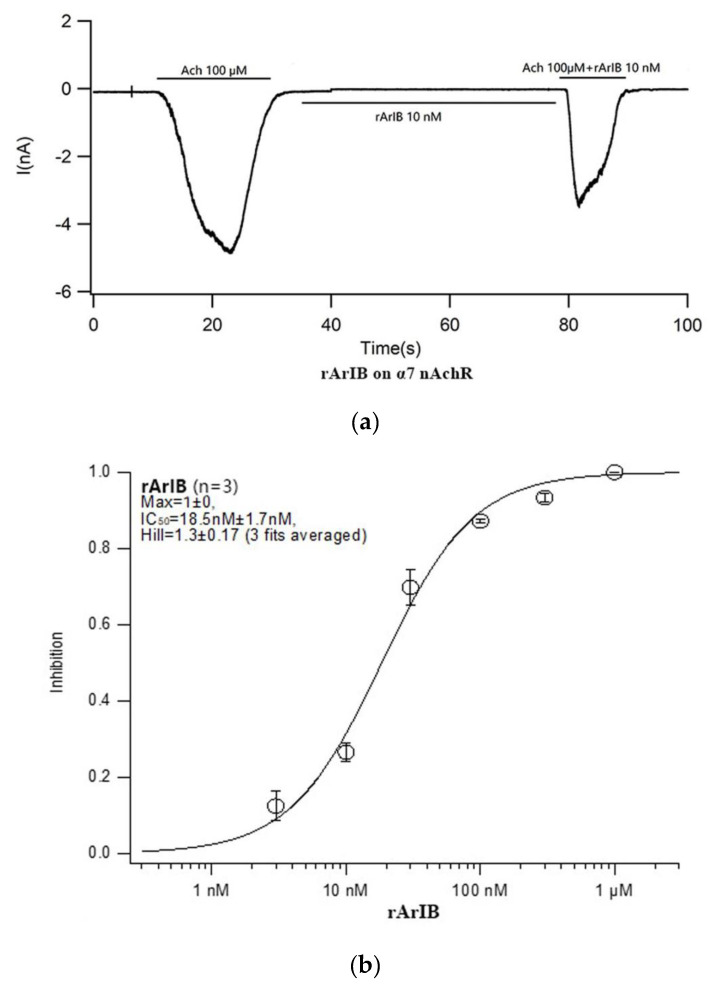
Analysis of Ach-induced current block by rArIB and rArIB (V11L, V16A). (**a**,**b**), rArIB blocks hα7 nAChR with an IC_50_ = 18.5 ± 1.7 nM, nH = 1.3 ± 0.7. (**c**,**d**), rArIB (V11L, V16A) blocks hα7 nAChR with an IC_50_ = 9.7 ± 0.4 nM, nH = 2.1 ± 0.055. No activity for rArIB (V11L, V16A) on hα9α10 and hα1β1δε nAChRs whereas (**e**,**f**). h—human; nH—Hill slope.

**Table 1 marinedrugs-18-00422-t001:** The synthetic gene sequences of ArIB and ArIB (V11L, V16A) with restriction enzyme digestion sites *Kpn* I and *Eco*R I. Introduced enterokinase site, restriction enzyme sites *Kpn* I and *Eco*R I were underlined.

Name	Sequence
ArIB Forward primer 1	5′-CGACGACGACGACAAGGATGAATGCTGTAGCAACCCGGCGTGCCGCGTGAACAATCCGCATGTTTGTCGTCGCCGTTAAG -3′
Reverse primer 1	5′-AATTCTTAACGGCGACGACAAACATGCGGATTGTTCACGCGGCACGCCGGGTTGCTACAGCATTCATCCTTGTCGTCGTCGTCGGTAC -3′
ArIB (V11L, V16A) Forward primer 2	5′-CGACGACGACGACAAGGATGAATGCTGTAGCAACCCGGCGTGCCGCCTGAACAATCCGCATGCATGTCGTCGCCGTTAAG-3′
Reverse primer 2	5′-AATTCTTAACGGCGACGACATGCATGCGGATTGTTCAGGCGGCACGCCGGGTTGCTACAGCATTCATCCTTGTCGTCGTCGTCGGTAC -3′

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
