# Peer review of "α-Conotoxin as Potential to α7-nAChR Recombinant Expressed in Escherichia coli"

_marinedrugs, 2020, doi:10.3390/md18080422_

Round 1

Reviewer 1 Report

A general observation is that the study contains new and interesting information on on a method to biosynthesize α-conotoxin ArIB and rArIB (V11L, V16A) by expression in Escherichia coli in an efficient and economical way. To this reviewer’s opinion, the research described is considered as of good quality and scientifically sound, the experiments are well-designed and the amount of data reported is sufficient, so there is only need for minor improvements which will make the manuscript more concise and reader friendly and also a need to improve the discussion section. On the other hand, however, the biggest problem of the manuscript is the use of English, which in many points compromises text comprehension and therefore needs very substantial improvement; there are numerous grammatical and syntax errors that need to be corrected upon revision, so the authors are kindly advised to have their manuscript checked by a native English speaker or a relevant professional service. Detailed comments follow:

General remark
- Please explain all abbreviations used at their first instance in the text (e.g. RP-HPLC, VGCC, GABAB GPCR and many others).
- Please place in italics all scientific names e.g. Escherichia coli, Xenopus laevis, Conus spp., etc. throughout the text (many instances).

Abstract
- Page 1, Lines 13-14: “However, soluble expression of functional peptides with multiple disulfides in Escherichia coli is still difficult.” This sentence is unconnected with the previous text. What do the authors mean? Please clarify as the syntax does not make sense.

1. Introduction
- Page 2, line 54: "proved” does not make sense. Maybe change to “which proved that” or “proving that”?
- Page 2, line 55: Please change “potently” to “potent”.
- Page 2, lines 50-62: This paragraph generally needs a lot of “polishing” with regard to English.
- Page 2, lines 67-68: “with lack the subtype”: what do the authors mean by this? (also used in page 8, line 204).
- Page 2, lines 69-71: Generally the syntax is confusing, impeding text comprehension, please rephrase.

2. Results:
- Page 4, lines 140-141: Were the retention times between different runs/ replicates so precisely the same to report the values with 2 decimal places (22.99 and 23.91) or was this the result of only one run? If not, please provide the average RTs (and standard deviations).
- Page 5, line 151: Please correct “Vaydac” to “Vydac”.
- Page 5, line 153: The wavelength for absorbance monitoring should also be mentioned in the Materials and Methods section (section 4.4), where the use of RP-HPLC is described.

3. Discussion:
- The discussion section is rather weak, as it generally continuing the results reporting and making very few comparisons or connections to other, already published, scientific works on the same topic. Please revise/reorganize the discussion, so that it is better substantiated.
- Page 8, lines 200-201: “Among them, α-conotoxin has been identified to be specifically pharmacological properties and target …”: Please rephrase, syntax is not making sense.
- Page 8, lines 205-207: “Conotoxins are usually prepared by solid-phase peptide synthesis, the limitation of which is not economical, time consuming, and not environmentally friendly, especially is the difficulty in synthesis of some peptide with more than 30 amino acids in length.”: Please rephrase, syntax is not making sense.
- Page 8, lines 214-216: “Using purified tags to be helpful for the purification of recombinant protein, such as glutathione S-transferase (GST), and hexahistidine (His6) etc.”: Please rephrase, syntax is not making sense (sentence without a verb).
- Page 8, line 219: “Yeast”: Which type of yeast? Also, this is a category and not a genus or species, therefore no italics should be used. If the authors intend to refer to yeasts in general, this should be in plural.
- Page 8, line 233: “effection”: do you mean “effect”?
- Page 8, lines 233-235: “As the fusion protein of Trx and His-tag would have tremendous effection on the structure and bioactivity of rArIB and rArIB (V11L, V16A), compared with other protease cleavage, SUMO, TEV, Factor Xa, and thrombin etc.,…”: Please use a reference to support the validity of this statement.

4. Materials and Methods:
- Page 9, line 276: The text mentions Table 1, but no such table is provided in the manuscript.

Author Response

Dear revewer:

Thanks for your hard and serious review! We have been revised, follwing as:

- Please explain all abbreviations used at their first instance in the text (e.g. RP-HPLC, VGCC, GABAB GPCR and many others).

A: accepted.

- Please place in italics all scientific names e.g. Escherichia coliXenopus laevisConus spp., etc. throughout the text (many instances).

A: accepted.

Abstract
- Page 1, Lines 13-14: “However, soluble expression of functional peptides with multiple disulfides in Escherichia coli is still difficult.” This sentence is unconnected with the previous text. What do the authors mean? Please clarify as the syntax does not make sense.

A: It is indeed a bit obtrusive and has been deleted.

  1. Introduction
    - Page 2, line 54: "proved” does not make sense. Maybe change to “which proved that” or “proving that”?

A: "proved” has been changed “also proving that”, line 56

- Page 2, line 55: Please change “potently” to “potent”.

A: “potently” has been changed “potent”, line 59

- Page 2, lines 50-62: This paragraph generally needs a lot of “polishing” with regard to English.

A: This paragraph has been polished, line 51-67

- Page 2, lines 67-68: “with lack the subtype”: what do the authors mean by this? (also used in page 8, line 204).

A: “with lack the subtype” means “It can be used as a new tool to distinguish α7 from other nAChRs subtypes”, the sentence has been revised, line 74

- Page 2, lines 69-71: Generally the syntax is confusing, impeding text comprehension, please rephrase.

A: This paragraph has been revised, line 78-82

  1. Results:
    - Page 4, lines 140-141: Were the retention times between different runs/ replicates so precisely the same to report the values with 2 decimal places (22.99 and 23.91) or was this the result of only one run? If not, please provide the average RTs (and standard deviations).

A: No, the retention times between different runs/ replicates were a little time shift, and related with the concentration of sample, we choose relative perfect images in the study. we has revised, line155,156

- Page 5, line 151: Please correct “Vaydac” to “Vydac”.

A: Vydac has been corrected, line 166

- Page 5, line 153: The wavelength for absorbance monitoring should also be mentioned in the Materials and Methods section (section 4.4), where the use of RP-HPLC is described.

A: “The detection wavelength is 214 nm” has been added in the Materials and Methods section (section 4.4), line 364

  1. Discussion:
    - The discussion section is rather weak, as it generally continuing the results reporting and making very few comparisons or connections to other, already published, scientific works on the same topic. Please revise/reorganize the discussion, so that it is better substantiated.

A: The discussion section has been reorganzied, line 202-303

- Page 8, lines 200-201: “Among them, α-conotoxin has been identified to be specifically pharmacological properties and target …”: Please rephrase, syntax is not making sense.

A: “Among them, α-conotoxin has been identified to be specifically pharmacological properties and target …” has been changed “Among them, α-conotoxin has been identified to be specifically pharmacological properties targeting a variety of nAChRs”, line 214-215

- Page 8, lines 205-207: “Conotoxins are usually prepared by solid-phase peptide synthesis, the limitation of which is not economical, time consuming, and not environmentally friendly, especially is the difficulty in synthesis of some peptide with more than 30 amino acids in length.”: Please rephrase, syntax is not making sense.

A: the sentence has been rivesed as “As so far, there are three major methods to obtain conotoxins: (1) seperation and extraction from cone snails; (2) solid-phase peptide synthesis; (3) recombinant expression in different expression system. Seperation and extraction conotoxins from cone snails are difficult to obtain in large quantities which could not meet the further research. There are great difficulties in solid-phase peptide synthesis for conotoxins with more than 30 amino acids in length, moreover, solid-phase peptide synthesis of conotoxins needs subsequent folding steps, which is not economical and time consuming.” Line 219-225.

- Page 8, lines 214-216: “Using purified tags to be helpful for the purification of recombinant protein, such as glutathione S-transferase (GST), and hexahistidine (His6) etc.”: Please rephrase, syntax is not making sense (sentence without a verb).

A: the sentence has been rivesed as “Using purified tags is helpful for the purification of recombinant protein, such as glutathione S-transferase (GST), and hexahistidine (His6) etc”, line 234-236.

- Page 8, line 219: “Yeast”: Which type of yeast? Also, this is a category and not a genus or species, therefore no italics should be used. If the authors intend to refer to yeasts in general, this should be in plural.

A: the type of yeast has been descibed as “the Yeast Pichia pastoris”, line 239.

- Page 8, line 233: “effection”: do you mean “effect”? 

A: “effection” has been changed “effect”, line 267

- Page 8, lines 233-235: “As the fusion protein of Trx and His-tag would have tremendous effection on the structure and bioactivity of rArIB and rArIB (V11L, V16A), compared with other protease cleavage, SUMO, TEV, Factor Xa, and thrombin etc.,…”: Please use a reference to support the validity of this statement.

A: a related reference has been incited, line 270

  1. Materials and Methods:
    - Page 9, line 276: The text mentions Table 1, but no such table is provided in the manuscript.

A: Table 1 has been inserted, line 102-104

Reviewer 2 Report

Revision Paper: α-conotoxin as potential to α7-nAChR recombinant expressed in Escherichia coli.

Yanli Liu, Yifeng Yin, Yunyang Song, Kang Wang, Fanghui Wu, Hui Jiang.

In this manuscript, the Liu and co-workers describe the study of procedures to overcome the challenge of soluble expression of functional peptides with multiple disulfides in Escherichia coli namely α-conotoxin ArIB and rArIB (V11L, V16A).  Aided with different techniques, from recombinant expression to, chromatographic hyphenated techniques and, electrophysiology to reach their goal. The study is quite interesting and very well conducted, with and undoubtedly importance to the scientific community and to the general public. This work draws attention also to another important issue, the need of rapid, efficient and economic biosynthesis techniques to obtain great amounts of disulfide-rich polypeptides, opening a door of possibilities regarding drug manipulation and their potential health application.

As an English non-native speaker, I understand the difficulties of writing in a language distinct from our mother tongue. I would recommend a minor English revision of the manuscript since it showed some small errors. The majority of the references are updated which is a value point.

Comments/Improvement corrections:

Only very few typing and abbreviation errors are present and are probably removed by the copyeditor, I give some examples:

Keywords: recombinant

Page 4 line 142: m/Z should be – m/z  (italic)

Page 5 line 151: “Vaydac”

Page 8 line 189: amino acids

Page 8 line 221: “…recombinant has been up to 100-500 mg/L” (space needed)

Page 8 line 225: “There are following…work:……”

Introduction Section

Along this section and throughout the whole MS authors should correct genus and species name to italic:

- Escherichia coli / E. coli

- Xenopus laevis

- Conus arenatus

Also, I find important to mention readers what type of organism Conus is, to pinpoint this toxin origin, thus also reinforcing the aim of this work and also adding it an ecological significance.

Results/Materials and Methods Section

Results are well presented, though I find pertinent that conditions of hyphenated techniques (RP-HPLC and MALDI-TOF MS) should be detailed and described in the MM section.

Legend of Figure 5 should be in the MM section.

Which grade were the reagents used in the chromatographic techniques? Please add this information in the MS.

Quality of Figure 7 should be improved.

Discussion

I recommend some amendments in this section:

Page 8 lines 188-191: “Conotoxins are a kind of small bioactive peptides ribosomally synthesized” … in which organisms? Please add a reference to support this sentence.

Figures shouldn’t be mentioned in the discussion section.

Also, I find some paragraphs more descriptive (ex: page 8 lines 233-252), resembling the results section, please re-formulate.

Author Response

Dear revewer:

Thanks for your hard and serious review! We have been revised, follwing as:

 Keywords: recombinant

 A: “recombinant” has been corrected, line 27

Page 4 line 142: m/Z should be – m/z  (italic)

 A:m/z (italic) has been corrected, line 156.

Page 5 line 151: “Vaydac”

 A: Vydac has been corrected, line 166

Page 8 line 189: amino acids

 A: amino acids has been corrected, line 203

Page 8 line 221: “…recombinant has been up to 100-500 mg/L” (space needed)

 A: accepted, line 259

Page 8 line 225: “There are following…work:……”

 A: accepted, line 245

Introduction Section

Along this section and throughout the whole MS authors should correct genus and species name to italic:

Escherichia coli / E. coli

- Xenopus laevis Xenopus laevis

Conus arenatus

  A: accepted

Also, I find important to mention readers what type of organism Conus is, to pinpoint this toxin origin, thus also reinforcing the aim of this work and also adding it an ecological significance.

Results/Materials and Methods Section

Results are well presented, though I find pertinent that conditions of hyphenated techniques (RP-HPLC and MALDI-TOF MS) should be detailed and described in the MM section.

 A: RP-HPLC and MALDI-TOF MS has been described in the MM section 4.4, line 356-367

Legend of Figure 5 should be in the MM section.

 A: Legend of Figure 5 only quick describe the RP-HPLC methods, detailed and described in the MM section 4.4, line 356-367

Which grade were the reagents used in the chromatographic techniques? Please add this information in the MS.

 A: accepted, in the MM section 4.1, Acetylcholine chloride (ACh) and other chemical reagents were all of analytical grade and purchased from Sigma-Aldrich (USA). Line 313-314

Quality of Figure 7 should be improved.

 A: accepted, the resolution has met magazine requirements, maybe the size of image was a little small which can be enlarged.

Discussion

I recommend some amendments in this section:

Page 8 lines 188-191: “Conotoxins are a kind of small bioactive peptides ribosomally synthesized” … in which organisms? Please add a reference to support this sentence.

A: a reference has been incited, line 203

Figures shouldn’t be mentioned in the discussion section.

A: accepted, figures has been deleted.

Also, I find some paragraphs more descriptive (ex: page 8 lines 233-252), resembling the results section, please re-formulate.

A: accepted, the discussion section has been reorganzied, line 202-303

Round 2

Reviewer 2 Report

Revision Paper: α-conotoxin as potential to α7-nAChR recombinant expressed in Escherichia coli.

Yanli Liu, Yifeng Yin, Yunyang Song, Kang Wang, Fanghui Wu, Hui Jiang.

In this manuscript, the Liu and co-workers describe the study of procedures to overcome the challenge of soluble expression of functional peptides with multiple disulfides in Escherichia coli namely α-conotoxin ArIB and rArIB (V11L, V16A).  Aided with different techniques, from recombinant expression to, chromatographic hyphenated techniques and, electrophysiology to reach their goal. The study is quite interesting and very well conducted, with and undoubtedly importance to the scientific community and to the general public. This work draws attention also to another important issue, the need of rapid, efficient and economic biosynthesis techniques to obtain great amounts of disulfide-rich polypeptides, opening a door of possibilities regarding drug manipulation and their potential health application.

I've already read this manuscript, regarding my queries/suggestions the authors response was satisfactory, though I still have some concerns that I will state below:

Introduction

I still find pertinent a small paragraph reinforcing the biology and ecology of cone snails as an endemic species, adding an ecological value to the performed work.

Results & Material and Methods Section

Figure 5: " with a Vydac"

Page 14:

I still have to insist, hyphenated methods are not described as they should, since they play an important role regarding purification and identification.

Apparatus models of both techniques are not displayed in the text, also MALDI-TOF MS conditions are not described, nor a reference to the purification method.

Author Response

Dear revewer:

Thanks for your hard and serious review! We have been revised, follwing as:

I still find pertinent a small paragraph reinforcing the biology and ecology of cone snails as an endemic species, adding an ecological value to the performed work.

A:done, line 80-81, 91-93

Figure 5: " with a Vydac"

A: done, line 168-169

Page 14:

I still have to insist, hyphenated methods are not described as they should, since they play an important role regarding purification and identification.

Apparatus models of both techniques are not displayed in the text, also MALDI-TOF MS conditions are not described, nor a reference to the purification method.

A: the purification method and MALDI-TOF MS conditions has been revised in the MM section 4.4 and 4.5, line 375-390.